# The Landscape of the Emergence of Life

**DOI:** 10.3390/life7020027

**Published:** 2017-06-16

**Authors:** Sohan Jheeta

**Affiliations:** Network of Researchers on Horizontal Gene Transfer and the Last Universal Common Ancestor (NoR HGT & LUCA), Leeds LS7 3RB, UK; sohan@sohanjheeta.com; Tel.: +44-0113-2628767

**Keywords:** DNA, RNA, ncRNA, preLUCA, LUCA, virus, virus connections, proteins, genetics first, metabolism first

## Abstract

This paper reports on the various nuances of the origins of life on Earth and highlights the latest findings in that arena as reported at the Network of Researchers on Horizontal Gene Transfer and the Last Universal Common Ancestor (NoR HGT and LUCA) which was held from the 3–4th November 2016 at the Open University, UK. Although the answers to the question of the origin of life on Earth will not be fathomable anytime soon, a wide variety of subject matter was able to be covered, ranging from examining what constitutes a LUCA, looking at viral connections and “from RNA to DNA”, i.e., could DNA have been formed simultaneously with RNA, rather than RNA first and then describing the emergence of DNA from RNA. Also discussed are proteins and the origins of genomes as well as various ideas that purport to explain the origin of life here on Earth and potentially further afield elsewhere on other planets.

## 1. Introduction

The time-frame for life on Earth to emerge was thought to have taken no more than 300 million years [1]; in terms of the geological age of the Universe at approximately 13.8 billion (10^9^) years, this is a relatively short period (~2.17%). The latest research suggests that, after a few 100 million years of accretion of the Solar System at approximately 4.67 billion years [2], life emerged on Earth somewhere between 4.3 and 3.8 billion years [3,4] and so the figure of 300 million years is not unreasonable. Even so, with our fairly good understanding of how life evolved on our planet (principally, Darwin’s theory of evolution by natural selection), our comprehension of its origin is still exceptionally limited. 

Most hypotheses of the origin and early chemical evolution of life focus on two well-trodden routes, either the “metabolism first” or the “genetics first” models. The former approach generally concentrates on the non-biological chemical reactions of metabolism that form the basis of extended pathways and cycles with the eventual emergence of genetics; the front runner in this case is called the “alkaline hydrothermal vent hypothesis” [5]. The genetic first (e.g., the “RNA World”) model assumes that metabolic ribozymes existed per se [6] and that these ribozymes played an instrumental role in the formation of the very first rudimentary peptides which took over the role of catalysis from RNA molecules, with the subsequent emergence of metabolism. From this it can be surmised there is chasm between these two hypotheses, in that in modern biology, both information and metabolism rely upon each other in order to make cellular biology function. The Network of Researchers on Horizontal Gene Transfer and the Last Universal Common Ancestor (NoR HGT and LUCA) was formed in 2013 and is attempting to bridge this gap by inviting researchers from various disciplines to share and disseminate their research and discoveries. This is an innovative and forward thinking group whose prime aim is to bring together researchers from a wide spectrum of scientific fields, including those who are not necessarily directly involved with the questions of the origin of life (e.g., medical virologists or researchers involved in food technology); the premise here is that with collaboration and cross pollination of ideas, this network may bring forth clues to the exact details of life’s first emergence, however tentative such clues may initially appear to be. In this paper, I give an account of some of latest discoveries, as conveyed at the 3rd NoR HGT and LUCA meeting held at the Open University, Milton Keynes on 3–4 of November 2016.

## 2. The Last Universal Common Ancestor (LUCA)

What is LUCA? It is a construct to explain the emergence of Archaea and Bacteria (and subsequently Eukarya). Gogarten (University of Connecticut, Mansfield, CT, USA) defined LUCA as an organism located at the deepest split in the Tree of Cells (ToC), between the bacterial domain on one side, and the archaeal domain and eukaryotic nucleocytoplasm on the other [7]. Analyses of molecular phylogenies reveal LUCA as an “almost” fully formed cellular organism possessing membranes used in chemiosmotic coupling; a DNA-based genetic repository; at least a rudimentary DNA replication system; transcription and ribosomal protein biosynthesis, including the ability to aminoacylate transfer RNAs using protein enzymes and messenger RNA directed peptide synthesis using 20 genetically encoded amino acids [8,9]. Predating LUCA by a long evolutionary history were progenotes, which represent an evolutionary phase when the molecular machineries for replication, transcription and translation were “*still in the process of evolving the relationship between genotype and phenotype*” [10]. The evolutionary path leading to LUCA was long and complicated. Reconstruction of ancestral aminoacyl-tRNA synthetases show that these enzymes, which are related to one another by ancient gene duplications, did not co-evolve with the genetic code, revealing a more ancient, possibly RNA based system to charge tRNAs with their cognate amino acids [11]. 

Horizontal gene transfer (HGT) was rampant during both the progenote and LUCA epochs. Therefore, the last universal common ancestors of the different proteins and RNAs did not all coexist in the cellular LUCA, rather these molecular most recent common ancestors existed in different lineages and at different times [12]. Representing evolutionary history as a single tree ignores the many reticulations that occurred throughout evolution [13]. Consideration of these reticulations and of extinction events reveals that, at the time of the organismal LUCA, several other organismal lineages existed [14,15], and that these lineages contributed genes to the present-day organisms [16].

Farias et al. (Universidade Federal da Paraiba, João Pessoa, Brazil) [17] reconstructed the ancestral peptidyl transferase centre from tRNA ancestral sequences (mentioned above). They suggest a model for the emergence of a primitive translation system, where a primitive ribosome worked as an adhesion centre resulting in the formation of a proto-tRNA; at this stage peptide synthesis occurred in the absence of a genetic code and with just the emergence, from anticodons, of initial genes, which were co-opted as part of the encoding system. Since LUCA is a theoretical construct to explain the possible routes to the understanding of the origin of the three domains of life, Prosdocimi et al. questioned its biological nature [18]. Farias et al. reconstructed the “best fit” molecular models of the ancestral sequences of tRNAs to test the hypothesis that these molecules constituted the initial genes [19]. Their results showed that the proteome, before LUCA (remembering that it is an almost cellular organism), may have been composed of basal energy metabolism, namely three carbon compounds, as in the glycolytic pathway, which operated as a substrates distribution centre for the development of metabolic pathways for nucleotides, lipids and amino acids. It was suggested that the assembly of metabolic pathways was taking place prior to the “fully-fledged” LUCA and that the resulting organisms could be construed as preLUCAs which also contained initial genes in the form of tRNAs and they developed the proteome/tRNAs theme further by analysing 400 protein sequences, concluding that the early carbon usage metabolic pathways worked as “binders of the substrates” thereby increasing the efficiency of this proto-enzyme machinery, which also involved the early activated tRNA~aa molecules. The outcome of these findings is the postulation of the presence of preLUCAs (virtually equivalent to the aforementioned progenotes). 

One approach to the understanding of the origin of life and its subsequent evolution is to construct a simple self-replication system and investigate the successive evolution—Mizuuchi (Osaka University, Osaka, Japan) did just that, by developing a translation-coupled RNA replication system. This was achieved by combining a reconstructed *E. coli* translation system and an artificial RNA genome encoding only a gene of RNA-dependent RNA-polymerase, Qβ replicase, which replicates the RNA genome once translated. Such a simple life-like system with only a small number of components and biological functions could be construed to resemble a molecular system of a preLUCA. These experiments demonstrated that a simple artificial self-replication system has a certain ability of limited adaptive evolution—for further information readers may wish to access the following papers [20,21].

Did LUCA or its predecessor preLUCA possess the ability to sense the physico-chemical parameters in the environment? Properties such as ambient temperature, pH, salinity, osmotic pressure, and presence of nutrients and poisons have been part and parcel of the environs of all cellular life since its emergence into the three domains. At what stage did the first cells evolve the ability to adjust to the changing environments? Galperin (The National Center for Biotechnology Information, Bethesda, MD, USA) addressed these points by studying the genomic distribution of signal transduction proteins. Of these, the two-component signalling systems (i.e., sensor histidine kinases and response regulators); methyl-accepting chemotaxis receptors (MCPs); diadenylate cyclases; Ser/Thr protein kinases and protein phosphatases are found both in Bacteria and Archaea. However, in Archaea, phylogenetic distribution of signal transduction systems is extremely biased, indicating that archaeal MCPs and at least some sensor histidine kinases have been acquired from Bacteria via HGT. Thus, only Ser/Thr protein kinases and protein phosphatases could be confidently traced back to LUCA, which could be due to the higher stability of phosphoSer and phosphoThr as compared to phosphoHis and phosphoAsp residues. Whether sensor histidine kinases and diadenylate cyclases were already present at the level of LUCA remains an open question [22].

## 3. Viral and Replicator Connections

Sorci (Università Politecnica delle Marche, Roma, Italy) put forward a notion that there are at least fifty phages encoding their own biosynthetic pathways yielding nicotinamide adenine dinucleotide (NAD) from vitamin precursors, namely nicotinamide (Nm) and nicotinamide riboside (NmR). These pathways are distinct from the bacterial host NAD biosynthesis, a promising source for novel antibacterial agents itself [23], as it is supported by a different set of viral genes. An even more numerous group of phages encode a unique NAD-related pathway, unprecedented in the highly-diversified synthesis of NAD across kingdoms [24], and possibly involved in the hi-jacking of host metabolism. Moreover, a phylogenetic analysis of the phage NAD biosynthetic shunt revealed a complex evolutionary scenario dominated by cross-kingdom HGT events. Why is this important? Since it is often cited that viruses had a parallel evolution to that of LUCA [25], viruses contributed to the fine-tuning of metabolic processes in the early evolutionary history of life. The cross-kingdom gene transfer events in cellular life forms is achieved via a mechanism known as transduction, one of the three known mechanisms of HGT. The question here is, did *such* gene transfers exist during the preLUCA epoch? This is a moot point to which we have no clear answers at present, although it is assumed that HGT of some type was present [26,27,28].

Soon after the emergence of self-replicating nucleic acids, like RNA (and later DNA), a simple preLUCA would have developed. It is likely that almost simultaneously, parasitic nucleic acid *replicators* would have emerged that could use these early examples of life in a similar way to viruses. Understanding these parasite-host dynamics is crucial to uncovering the emergence and evolution of life, and contemporary viruses can serve as a useful model as proposed by Aswad (University of Oxford, Oxford, UK). Although viruses do not leave behind a fossil record, we can use techniques from paleovirology to determine long-term evolutionary dynamics of virus-host interactions [29]. Central to this approach is taking advantage of the large scale genomic data collection. Specifically, endogenous viruses can be found in the genomes of their hosts, having integrated millions of years previously [30,31]. As well as using such ancient viruses as a window into their natural history, techniques in paleovirology integrate methods from metagenomics that allow the discovery of novel viruses (both ancient and extant). Using such methodologies, Aswad was able to discover fifteen new viruses that belong to a new lineage of large DNA viruses. Moreover, he reported on how methods from paleovirology can be used to uncover the details of complex horizontal gene transfer events both between viruses as well as multi-laterally between host, viruses and satellite viruses [32,33]. Details on how the evolutionary dynamics of both host and viruses are influenced by such gene exchanges are beginning to emerge in a series of recent studies, such as the regulation of an anti-viral defence gene in mice by an integrated retrovirus [34].

Tuller (University of Tel Aviv, Tel Aviv, Israel) used statistical modelling to study the way viruses’ genomes encode their replication efficiency, which could elucidate the understanding of the mechanisms of HGT in connection with the origin of life. A natural research question posed was, how are various viral gene expression codes and the viral life cycle related? To attempt to answer this, his group analysed dozens of viruses (e.g., dsRNA, ssRNA, dsDNA, ssDNA, etc.), and numerous hosts (e.g., vertebrates, bacteria, fungi, etc.) and concluded that various “hidden” regulatory signals are encoded in the viral genome which naturally affect their evolution and replication rate. Codes analogous to the viral “hidden” codes may have appeared in the molecules prior to the origin of cellular life [35,36,37,38].

## 4. Metabolism First

Fox (University of Hohenheim, Stuttgart, Germany) contended that prebiotic life could have been kick-started at the foot of volcanoes, in small rock-pools of water beginning with the ‘proto-metabolism first’ rather than the genetics first hypothesis. A simple premise here is that such a small volumetric body of water would have aided in concentrating the necessary proto-metabolism molecules due to constant evaporation/rehydration cycles. Conversely in open ocean waters, due to the presence of large volumes of water, especially at the hydrothermal vents, dilution of the initial prebiotic molecules would have been problematic. In addition, condensation reactions during polymerisation of amino acids, nucleotides and precursors of lipids would have been equally challenging as a water molecule would have been generated during each and every reaction. Fox showed that a series of experiments simulating the thermal alteration of amino acids in rock-pools resulted in the formation of pyrroles; subsequent reactions between these pyrroles brought about the synthesis of porphyrins. It is further postulated that during the Hadean epoch, in the presence of suitable “Hadean-minerals”, metalloporphyrins could be formed. The last products are exceptionally important in that they can transport electrons and harness light, an inexhaustible supply of energy; in modern organisms, these functions are also carried out by cytochromes and chlorophylls respectively. Moreover, porphyrin-type cofactors could have played an important role in the chemical evolution of life by aiding in the formation of important molecules such as peptides and RNA and then eventually DNA [39]. The central tenet of the metabolism first hypothesis is that, initially, protoenzymes were made of amino acids. Iqubal (Indian Institute of Technology, Roorkee) proposed that it is possible to carry out condensation of amino acids on the surfaces of various mineral surfaces. Minerals were part and parcel of the natural inventory of early Earth’s early lithosphere. He showed that oligomerisation of simple amino acids like glycine and alanine could take place on the surface of nickel ferrite (NiFe_2_O_4_), cobalt ferrite (CoFe_2_O_4_), copper ferrite (CuFe_2_O_4_), zinc ferrite (ZnFe_2_O_4_) and manganese ferrite (MnFe_2_O_4_) nanoparticles in the temperature range of 50–120 °C during the duration of 1–35 days without applying any wetting/drying cycles. Among the metal ferrites tested for their catalytic activity, nickel ferrite produced the highest yield of alanine dimer and glycine trimer [40]. The idea here is that given time, larger oligomers with rudimentary catalytic activities could be made relatively easily. Such oligomers may then participate in various catalysed reactions leading to more evolved pathways, cycles, hypercycles with genetics following on from these. According to Iqubal and co-workers, spinel metal ferrites might have played an important role in the context of prebiotic chemistry and the origin of life by concentrating and further condensing the important “bio-monomers” in the primeval soup. The latest studies involving spinel metal ferrite and ribonucleotides result in their oligomerization [41,42], shedding light on the importance of spinel metal ferrites during the prebiotic chemistry epoch.

DEL GAUDIO (Università degli Studi di Napoli Federico II, Napoli, Italy) put forward an ambitious and novel proposition as to how life began on Earth. The central tenet of her approach is that both terrestrial rocks containing iron (II, III) oxide (e.g., serpentine, olivine) and some chondritic meteorite samples (e.g., ipertenic, Mocs and Holbrook, Finnmarken pallasite and siderite octahedrite) exhibited catalytic reactions [43]. The former rocks were assumed to be relics belonging to the prebiotic epoch and are now found to be common in present day Earth. These rocks under sterile conditions mediated a variety of reactions that are generally carried out by biological catalysts (enzymes) such as oxidoreductases, transferases, hydrolases, lyases, isomerases and ligases. From her observations, she put forward a “Multiple Root Genesis” (MuRoGe) hypothesis. This hypothesis predicts that life on Earth arose via multiple routes—for example both in situ on the Earth and also that it was made elsewhere in the Universe and then delivered onto the Earth (i.e., panspermia).

Hansma (University of California at Santa Barbara, Santa Barbara, CA, USA) observes that the spaces between mica sheets are a good “container” which could have brought together the reactants for chemical reactions needed for life’s origins. Mica sheets form sandwiches of sheets (mica “books”) with anionic crystal lattices bridged by potassium ions (K^+^). Mica’s crystal lattices have a lattice spacing of 0.5 nm, which is equal to the distance between anionic phosphate (PO_4_)^3−^ groups in RNA (and later DNA). Therefore, the K^+^ ion rich sheets, with their 0.5 nm spacing, provided a sort of primitive cell where chemical reactions could have occurred. The energy for chemical reactions would have been generated by the opening-and-shutting movements of mica sheets of the sandwich in response to the environmental temperature changes and fluid flow dynamics pertaining to ebb and flow. It was assumed that the role played by mechanical energy would eventually be taken over by ATP; how this was to be achieved is unclear, however. Moreover, mica sheets are hypothesised to be equivalent to membrane-less organelles, which preceded membrane-bound ones at the origins of life. These organelles (e.g., nucleoli) are thus relics of life’s chemical evolutionary beginnings [44,45,46].

Unlike Hansma’s first living entities in the form of mica sheets, the essence of Battaglia’s (University College London, London, UK) science is self-assembly of amphiphilic molecules, which could be positional in that it creates energy gradients by enclosing chemicals into aqueous volumes using gated compartments; the second aspect of self-assembly is ubiquitous in nature, forming the core of many biological transformations. Therefore, both compartmentalisation and positional self-assembly create structures, namely polymersomes, whose surfaces express several chemistries (e.g., chemical potentials across the membranes) performing their function holistically according to specific topological interactions. His research focused mainly on the precision control of thickness, brush density, mechanical properties, and permeability of amphiphilic block copolymers in water forming polymersomes with radii of 5 µm and upwards, giving a unique way in which the very first cell may have arisen [47,48].

## 5. Nucleic Acids and Protein Systems

Which polymer, RNA or DNA, came first and thus kick started the chemical origin of life? Historically, RNA is thought to predate DNA, simply because RNA can carry out, albeit slowly, limited specific types of catalytic activity as well as act as a carrier of chemical information in the form of genetic codes; also, noting that C2 in ribose in DNA is deoxy which requires some sort of enzyme to remove the highly electronegative [O], thereby positioning RNA as the more ancient molecule. As a result of RNA’s two “activities”, DNA is relegated to second place, meaning that it would have been made from the RNA which predated it. Krishnamurthy (The Scripps Research Institute, La Jolla, CA, USA) and his team thought otherwise. Their counter-intuitive argument runs along the lines that a transition of homogeneous RNA → homogeneous DNA seems unrealistic in the absence of separating and editing mechanisms in a prebiotic context. Therefore, the simultaneous presence of RNA and DNA molecules with the pre-existing alternative building structures (XNA, where X = unknown nitrogenous base); alternative sugars; alternative linker units and alternative recognition elements would have removed the need for “clean reactions” with high chiral purity demand during homogeneous → homogeneous transition. In the milieu of the prebiotic epoch, gradual chemical evolution of heterogeneous → homogeneous during the oligomeric/polymeric transitions led to the eventual emergence of homogeneous backbones of RNA:DNA → DNA/protein world as we recognise today. The net result being that it was a “chimeric oligonucleotide system” from which life emerged rather than a “pure” RNA world [49,50].

If there is life elsewhere in the universe, will it also use DNA? This was the question put forward by Devine (London Metropolitan University, London, UK) at the beginning of his oral presentation. In order to answer this point, he proceeded to examine the synthetic analogues of nucleic acids which differ from their natural counterparts in three key areas. These being (a) replacement of the phosphate moiety with an uncharged analogue; (b) replacement of the pentose sugars, ribose and deoxyribose, with an alternative pentose and hexose derivatives; and finally (c) replacement of the two heterocyclic base pairs adenine/thymine and guanine/cytosine with non-standard analogues that obey the Watson-Crick pairing rules. The conclusion was that a polyphosphate backbone was crucial to generating viable genetic biomolecules; sugar moiety can vary to some degree; and extra-terrestrials may not necessarily have DNA, but will have a sugar-phosphate backbone in their nucleic acid [51]. 

In presenting “the origin and early evolution of information transfer in biological systems”, Pohorille (NASA Ames Research Center, Mountain View, CA, USA) explored the question of whether linear, genomic information transfer might have been preceded by a simpler, non-linear information transfer system that operated at the origin of life. Based on both experimental and theoretical studies he advanced a thesis that the earliest information transfer was based on molecular self-assembly and recognition. Darwinian evolution could have progressed without a genome, leading to the formation of simple metabolic networks. However, it encountered barriers. The requirement for increasing efficiency, diversity and specificity created an imperative for the emergence of the modern, more precise linear information storage and transfer system. Thus, the genome itself was a product of evolution. Next, Pohorille discussed the evolution of simple genomes, focusing on questions such as: was the outcome of early evolution predictable or was it, instead, governed by chance? In addition, what was the role of “neutral mutations” in the evolution of increasingly complex systems? To address these questions, he and his collaborators explored fitness landscapes of RNA ligases that differed in length. The results support the existence of near-neutral networks and possibilities for evolutionary optimisation. Also, their findings, based on combined phylogenetic and structural data, suggest that complex modern RNA structures evolved from simple shared structures that were present in much shorter, although less efficient RNA molecules [52,53].

## 6. Hypothesis: ncRNA-Cellular Activity Controller?

RNAs are widespread in all biological systems (except for DNA viruses) and are involved in multi-laterally adapted systems that control numerous cellular processes, the magnitudes of which are still being explored. Principally, there are two broad categories of RNAs, namely coding and non-coding (ncRNA)—this summary refers only to the latter. 

ncRNA molecules, such as housekeeping RNAs (ribosomal, transfer, small nuclear, and small nucleolar RNAs) and the thousands of regulatory RNAs that are the subject of ongoing intense studies, can form structures ranging from primary to quaternary levels. These are as follows: primary structures of approximately 22 nucleotides, as in guided single stranded microRNAs (miRNA); double stranded miRNA interference segments exist as secondary shapes; tertiary level structures which are broadly composed of RNAs and protein complexes called ribonucleoproteins as in RNase P and RNase MRP (Mitochondrial RNA Processing) complexes; at the quaternary level, again in conjunction with proteins, various co-factors and metal ions (e.g., Zn), RNAs form huge ncRNA-nanomachines as in spliceosomes, the Varkud satellite (VS), RNA-induced silencing complex (RISC) and ribosomes. These structures are multifunctional and are broadly regulatory, being involved in gene regulation as well as interfering with and processing both small and large RNAs [54]. Such processing actions are well orchestrated, even to the point of efficient shredding of any unwanted RNAs—for example “used” coding mRNA within the cell is degraded rapidly (via RISC centres) so as to prevent them from being translated further. Recent discoveries have also demonstrated that ncRNAs can act as riboswitches (e.g., *glmS* ribozymes), whereby they regulate their own activity and perform genetic control by a metabolite binding mRNA [55]. They can control the activities of some plasmids (e.g., R1 plasmid of *E. coli*) via antisense RNA as in the *hok/sok* system [56]. It is suggested by Kotakis (also as outlined later) that ncRNA-directed transfer of genes can occur from “organellar” entities (e.g., chloroplasts) to the nuclei of eukaryotes [57]. Furthermore, ncRNAs can act as shielding triggers against invading mobile genetic elements, thereby affording protection against incoming attacks by any “parasitic” nucleotide sequences found in the environment [58]. ncRNAs, in addition to their ribozymatic activities and ability to carry genetic codes (e.g., influenza, an RNA virus), are significant in that the hallmark of their modular architectural structure implies that structural and possible functional similarities exist among them [54]. A unique aspect of ncRNAs is that they are highly conserved and so it is thought that they are molecular relics which delineated a “hypothetical” entity called LUCA, which pre-dated the three domains of life, namely Archaea, Bacteria and Eukarya. 

The conserved nature of ncRNAs allowed Jheeta (Chairman, NoR HGT and LUCA) to postulate that it is highly feasible that these ncRNA molecules could still have overall control of cellular activity and perhaps this is the reason why DNA replication still requires this ncRNA primer. This is particularly relevant as there are large numbers of newly discovered ncRNAs whose functions are still to be explained and validated. 

## 7. Horizontal Gene Transfer

Horizontal gene transfer (HGT) was a pre-requisite during the LUCA epoch and probably during the preLUCA era as well. It certainly continues to be widespread in cellular life forms, particularly in prokaryotes, on Earth today. HGT is important in that it is a necessary source of the genetic innovations occurring throughout the history of cellular life forms. Such innovations could be achieved via three main mechanisms, namely conjugation (plasmid-mediated), transduction (virus-mediated) and transformation (naked DNA uptake directly). Of these, the two former mechanisms resemble “infections”, whereas transformation is a highly-evolved feature of the recipient cell. The term HGT encompasses all three mechanisms despite their intrinsic differences and evolutionary consequences which were reported in the 2014 NoR HGT and LUCA meeting [28]. During the 2016 meeting, Ambur (Oslo and Akershus University College, Oslo, Norway) developed the importance of transformation further. He asserted that this mechanism is akin to eukaryotic sex and may therefore share selective advantages. During the studies of small DNA uptake sequences (DUS) in *Neisseriaceae* family [59] he revealed a conservative transformation output traceable in deep time [60]. These and other observations were discussed to generate an important nuance in the commonly held view that HGT brings about novelties and innovations.

It is difficult to specify an exact point as to when eukaryote cells emerged in the time frame of the evolution of life on the Earth, but it can be surmised that there are eukaryotes that existed at least 1 billion years ago, as testified by the triploblastic animal fossil evidence [61]. However, we know for certain that there was a merger between two prokaryotes that brought about the emergence of a “new” type of species as encompassed in the “endosymbiont” hypothesis [62]. In this scenario, one small cell resides in another larger cell, the small cell furnishes the larger one with a plentiful supply of energy while the larger cell reciprocates by providing the smaller cell protection; with the resultant entity being a eukaryote. Therefore, over the past 100’s of millions of years the smaller (i.e., mitochondria) cell’s genes (>1000) are transferred to the nucleus of the larger cell, leaving behind a mere 37 genes coding for 24 non-coding RNAs (namely 2 rRNAs and 22 tRNAs) and 13 coding RNAs for proteins belonging to the electron transport and oxidative phosphorylation system. These 37 genes are denoted as mitochondrial DNA (mtDNA) and are highly conserved, having remained largely unchanged for over 500 million years. Overballe-Petersen (University of Copenhagen, Copenhagen, Denmark) asked: why do eukaryotes still harbour mtDNA? The four hypotheses often discussed include: Genetic Code Disparity; Hydrophobic Constraint; Oxidative state sensing (CoRR, Co-location of Redox Regulation); and Remnants (i.e., full transfer has not yet occurred). Overballe-Petersen offered an alternate hypothesis entitled: “the proton pump hypothesis”. Based on protein structures and functional gene-transfers he proposed that proton-channels that disrupt cellular integrity may be the reason why mtDNA genes are prevented from transferring to the nucleus and therefore persist in mitochondria [63]. A test for this hypothesis would be to transfer the mtDNA ORFs (open reading frames) to the nuclear genome under a tightly repressed promoter as in a yeast strain, for example. This line of reasoning is the basis for his future research in relation to his hypothesis. 

Gene loss from organellar genomes to the nucleus has occurred throughout the evolutionary history of eukaryotes. The transfer of such genes to the nucleus is mainly governed by DNA-directed mechanisms; however, ncRNA-directed movements of genetic material to the nucleus have also been uncovered. This is particularly true of chloroplasts. Recent technical advances in organellar biotechnology, genome engineering and single-molecule tracking have given some novel insights into ncRNA chemistry at both cellular and organismal survival levels. From this research, Kotakis (The Hungarian Academy of Sciences, Budapest, Hungary) posited that an ncRNA-directed route in Archaea, although a rare occurrence, could have contributed to endosymbiotic gene transfer under certain micro-environmental conditions-encouraging “photosynthetic” genomes to migrate to the nucleus during the process of endobiosynthesis. It was proposed that ncRNA molecules with a particular structural configuration and respective functional attributes can drive evolution by reacting to environmental pressures while drowning-out epigenetic aberrations. The structural and functional aspects of ncRNA also affect redox and genetic characters leading to co-evolution. He surmised that RNA could substitute every step in the dogma of molecular biology concerning the flow of genetic information [57,64].

## 8. Life Elsewhere in the Universe

It is without a doubt that life exists on Earth, but what is not known is whether there is any life in our solar system (e.g., on Mars) or even further afield in the Universe, as none has been discovered to date. Shepherd (University of Arkansas, Fayetteville, AR, USA) investigated the potential of this issue by working with extremophiles; hardy microbes that exhibit a high potential to flourish in hostile environments such as −50 °C temperatures, pressures of 0.006 bar, and low water activity, among other stressors, which are all representative of the conditions on present-day Mars [65]. She studied the survival rates and biosignatures of sulfate-reducing bacteria (*Desulfotalea psychrophila*, *Desulfotomaculum arcticum*, and *Desulfuromusa ferrireducens*) which all demonstrate highly efficient survival mechanisms. After preliminary tests, growth was observed in each of the experiments for all three organisms; further DNA purification and isolation has led to successful PCR analyses for future DNA amplification. She concluded that such research is important not only for improved understanding of biology in harsh environments, but also to aid NASA’s planetary protection policy by assisting in the recognition and mitigation of potential contamination on extra-terrestrial bodies [66]. Such contamination would undoubtedly arise due to continued space missions.

The recent discovery of a large number of exoplanets raises a question: where does a newly-discovered exoplanet stand in its capability to develop life as we know it on Earth? Maccone’s (IAA SETI Permanent Committee, Italy; Istituto Nazionale di Astrofisica, Italy) answer to this question is to describe recent developments in a new statistical theory describing Evolution and SETI by mathematical equations. His theory deals with the mathematics of lognormal stochastic processes and he capitalises on the basic numbers involved with evolution: the time when life first appeared on Earth (3.5 billion years ago?); the number of species living on Earth nowadays (50 million or more?); and the Shannon Entropy to quantify molecular complexity between the first species (RNA organism?) and today’s humans, resulting in an overall molecular complexity increase of 25.575 bits/individual if the growth was exponential in the time, and 12.074 bits/individual if the growth was of Markov-Korotayev type (a cubic in the time) [67,68]. 

## 9. What’s Next?

Are we any nearer to obtaining a complete answer to the pertinent question of the origin of life? The information disseminated at this latest meeting basically yields up a miniscule yes, but a humungous no. On a positive note, at every NoR HGT and LUCA meeting something new is brought to the table. For example, Krishnamurthy’s announcement that RNA and DNA may have developed simultaneously from XNA, as well as Jheeta’s hypothesis that RNA may still be in overall control of cellular activities and, also, the assertion by Kotakis that RNA can control the movement of genes from an organelle to a nucleus—a question here is what else is RNA capable of doing? These are still early days in the quest for a total understanding of the exact mechanisms of life’s emergence on Earth; however, it is expected that with the continuance of such meetings as NoR HGT and LUCA we can collectively begin to make serious headway.

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
