# Peer review of "The Landscape of the Emergence of Life"

_life, 2017, doi:10.3390/life7020027_

Round 1

Reviewer 1 Report

In this conference report, Jheeta provides a detailed account of the latest achievements in the challenging field of the origin of life, as communicated by researchers that participated at 3rd NoR HGT & LUCA meeting held at the Open University, Milton Keynes on 3rd-4th of November 2016.

Besides a widespread consensus as to when life on Earth emerged (between 4.3 and 3.8 billion years ago), scientists still fumble around in the dark when asked “how did it happen?”. In an attempt to tackle one of the most fascinating science’s misteries, the Network of Researchers on Horizontal Gene Transfer and the Last Universal Common Ancestor (NoR HGT & LUCA) gathers multifaceted scientific expertise also encompassing researchers that are not full-time dedicated to such cause. This latter feature is an added value of this group of “origins of life hunters” managing to make consistent progress and bringing new valuable hypotheses at each meeting.

I read this report several times without finding any major flaw. It is well-written and the author excellently glued together all the contributions finding connections among apparently distant approaches and results. I recommend prompt publication of this report, but not without having preliminarly addressed the following minor comments.

Minor comments:

Throughout the test, the latin expression exempli gratia (for example) and id est (that is) should be consistently abbreviated as e.g. and i.e. instead of eg and ie. For instance: line 22: (eg the RNA world); line 192 (ie panspermia); line 278 (eg glmS ribozymes), and so on.

Line 42: replace “Eukaraya” with Eukarya.

Line 109: remove “and” in “pathways and yielding”.

Lines 110-1: accordingly, “This pathway” should be corrected to “These pathways”.

Lines 119-120: Add the question mark at the end of the sentence.

Lines 166: “The central tenet of the metabolism first hypothesis is that enzymes were made initially”: the meaning of this sentence is not clear; please rephrase.

Line 267-68: abbreviation for microRNA, i.e. miRNA should be introduced when first cited. Remove “(miRNA)” in line 268, which is a repetition anyways and place it in line 267.

 Line 271: conversely, punt ncRNA in brackets after the full word…. non coding RNA.

 Line 278: the semicolon is not necessary.

 Line 300: I suggest to modify “(plasmid)” and “(virus) into (plasmid-mediated) and (virus-mediated) for clarity.

 Line 302: “encompass” must be “encompasses”.

 Line 309: change “open frame reading” into “open reading frames”. Also, fix the typo “promotor” with “promoter”.

 Line 379: in the sentence” ..what else is RNA is capable of doing?” remove the extra “is”.

Author Response

Complied and checked text as per recommended

Reviewer 2 Report

Manuscript ID: life-198596

Type of manuscript: Conference Report

Title: The landscape of the emergence of life

Authors: Sohan Jheeta *

Referee report:

Sohan Jheeta reports about the 3rd NoR HGT & LUCA meeting held at the Open University, Milton Keynes, on 3rd–4th of November 2016. The manuscript starts by outlining concisely the scientific background of the meeting. Subsequently, the author presents the contributions to the meeting, sorted by topics.

The manuscript provides an interesting overview on the different fields of research which are related to the subject of this meeting. Because of the scientific diversity of the contributions, it is a challenging task to assess them all. As far as I can judge, the contributions are reproduced correctly and interpreted as it was envisaged by the presenters. Precisely because of the broadness of the topics, the manuscript is interesting and will attract a wide readership. I have only a few suggestions and corrections:

line 13:

...life was formed approximately at 4.67 billon years [2]...”: Bouvier and Wadhwa reported the age of the solar system and not the emergence of life. The author should revise this sentence.

lines 161–163:

There is a difference between light absorption and electron transport. Therefore, there is a difference in the functionality of chlorophylls and cytochromes. A corrected version of this sentence could read “The last products are exceptionally important in that they can transport electrons and harness light, an inexhaustible supply of energy - these functions are also carried out by cytochromes and chlorophylls, respectively, in modern organisms.“

lines 166–167:

The basic idea of the metabolism first hypothesis is catalysis, where the compound class of oligopeptides is assumed to be only one among many other catalysts. Therefore, and because enzymes are ribosomal products, I would not call the initial catalyst “enzymes.” It is perhaps more appropriate to write write “... that protoenzymes were made initially; protoenzymes are ...”

In summary, this inspiring manuscript is understandable and an important overview on recent origin-of-life related research activities. It has been carefully prepared. After minor revisions (see above), the manuscript will be perfectly suited for publication in the MDPI journal life.

Author Response

Complied and checked as per recommended

Reviewer 3 Report

line 89 and 92 replace pre-LUCA instead of preLUCA

line 60, at the end of first period (...that occurred throughout evolution.)  I suggest to evaluate additional reference  M. C. Rivera & J.A. Lake (2004) Nature 431, 152-155.

line 62 at the end of the sentence period, I suggest to evaluate an additional most recent reference as well as entitled " Horizontal gene transfer: building the web of life" publihed in 2015 in Nature Reviews Genetics vol 16,472–482

line 280 replace  "E.coli" instead of E.Coli

line 312 "......It is difficult to specify an exact point as to when eukaryote cells emerged in the time frame of the evolution of life on the Earth, but it can be safely stated that there are no eukaryotes older than 700 312 million years, as testified by the lack of fossil evidence"              my suggestion is to take into consideration adding as reference also the paper entitled "Triploblastic Animals more than 1 billion Years ago:trace fossil Evidence from India" pubblished  in 1998 in volume 282 of the journal Science  from page 80-83.

line 266  I suggest to edit by inserting to improuve the specication  "ncRNA,  such as housekeeping RNAs (ribosomal, transfer, small nuclear, and small nucleolar RNAs) and the thousands of regulatory RNAs that are the subject of ongoing intense studies, can form structures ranging from primary to quaternary levels, .......

Author Response

Complied and checked as per recommended